Future sea-level rise drives rocky intertidal habitat loss and benthic community change

Kaplanis Nikolas J. nkaplanis@ucsc.edu 1 2
Edwards Clinton B. 2
Eynaud Yoan 2
Smith Jennifer E. 2
1 Department of Ecology and Evolutionary Biology, University of California, Santa Cruz , Santa Cruz , CA , United States of America
2 Center for Marine Biodiversity and Conservation, Scripps Institution of Oceanography, University of California, San Diego , La Jolla , CA , United States of America
de Rezende Carlos
Electronic publication date: 2020 May 29
Publication date: 2020
Volume: 8
Electronic Location ID: e9186
Received 2019 Sep 6; Accepted 2020 Apr 23
Copyright: ©2020 Kaplanis et al.
Copyright year: 2020
Copyright holder: Kaplanis et al.
License: This is an open access article distributed under the terms of the Creative Commons Attribution License, which permits unrestricted use, distribution, reproduction and adaptation in any medium and for any purpose provided that it is properly attributed. For attribution, the original author(s), title, publication source (PeerJ) and either DOI or URL of the article must be cited.
License URL: https://creativecommons.org/licenses/by/4.0/

Keywords: Sea-level rise, Rocky intertidal, Habitat loss, Photogrammetry, Structure-from-motion, Remote sensing, Climate change, Large-area imaging, LiDAR

Funding: San Diego Foundation, through a Blasker Environment Grant National Science Foundation Graduate Research Fellowship Program NSF DGE 1842400 This work was made possible through the support of the San Diego Foundation, through a Blasker Environment Grant awarded to Nikolas J. Kaplanis, Clinton B. Edwards, and Jennifer E. Smith. This material is based upon work supported by the National Science Foundation Graduate Research Fellowship Program, awarded to Nikolas J. Kaplanis, under Grant No. NSF DGE 1842400. The funders had no role in study design, data collection and analysis, decision to publish, or preparation of the manuscript.

==============================
The impacts of sea-level rise (SLR) are likely to be the greatest for ecosystems that exist at the land-sea interface, where small changes in sea-level could result in drastic changes in habitat availability. Rocky intertidal ecosystems possess a number of characteristics which make them highly vulnerable to changes in sea-level, yet our understanding of potential community-scale responses to future SLR scenarios is limited. Combining remote-sensing with in-situ large-area imaging, we quantified habitat extent and characterized the biological community at two rocky intertidal study locations in California, USA. We then used a model-based approach to estimate how a range of SLR scenarios would affect total habitat area, areal extent of dominant benthic space occupiers, and numerical abundance of invertebrates. Our results suggest that SLR will reduce total available rocky intertidal habitat area at our study locations, leading to an overall decrease in areal extent of dominant benthic space occupiers, and a reduction in invertebrate abundances. As large-scale environmental changes, such as SLR, accelerate in the next century, more extensive spatially explicit monitoring at ecologically relevant scales will be needed to visualize and quantify their impacts to biological systems.

Introduction

Sea-level rise projections and potential impacts to the rocky intertidal zone

Sea-level rise is predicted to alter habitat availability and modify community structure in many marine ecosystems (Hoegh-Guldberg & Bruno, 2010; Nicholls & Cazenave, 2010). Long-term monitoring of sea-level shows considerable variability among locations, but an overall rising trend with recently increased rates of change at many locations (Church & White, 2006; Mcleod et al., 2010; Nicholls & Cazenave, 2010; Rahmstorf, 2010; National Research Council, 2012; Church et al., 2013; Cazenave et al., 2014; Chen et al., 2017). While estimates remain uncertain, recent studies project up to 2.5 m of SLR within the next century (Church et al., 2013; Sweet et al., 2017). Such a large magnitude and rapid shift poses a substantial risk to the integrity of coastal ecosystems, yet the extent to which SLR will modify the physical and ecological structure of rocky coastlines remains mostly unknown (Denny & Paine, 1998; Thompson, Crowe & Hawkins, 2002; Harley et al., 2006; Helmuth et al., 2006).

Certain characteristics of rocky intertidal systems make them particularly vulnerable to SLR driven habitat loss (Denny & Paine, 1998; Thompson, Crowe & Hawkins, 2002; Harley et al., 2006; Helmuth et al., 2006). When backed by steep cliffs or anthropogenic structures, as is the case on the majority of coastlines globally (Emery & Kuhn, 1982), many rocky shores are expected to experience “coastal squeeze”—a general narrowing of the extent of the intertidal zone (Doody, 2013; Pontee, 2013) and a steepening of the coastal profile (Vaselli et al., 2008). Whether SLR will lead to losses or gains in habitat at a given site is highly dependent upon geomorphology, rates of erosion, underlying stratigraphy, and alteration in sedimentation processes at local and regional scales (Emery & Kuhn, 1980; Young & Ashford, 2006; Masselink et al., 2020; Schaefer et al., 2020). Studies investigating durable coastlines with low erosion rates have suggested that SLR may cause substantial rocky intertidal habitat loss. Estimates of habitat loss range from 10–27% and 26–50% with 0.30 m and 1.90 m of SLR, respectively, in northern Scotland (Jackson & Mcilvenny, 2011), to 10% and 57% with 1.0 and 2.0 m of SLR, respectively, in Oregon, USA (Hollenbeck, Olsen & Haig, 2014). Thorner, Kumar & Smith (2014) found that with 0.3–1.0 m of SLR on five rocky headlands in New South Wales, Australia, impacts will be variable but will largely result in substantial habitat loss. Most recently, Schaefer et al. (2020) suggested that along 50 km of rocky shoreline in eastern Australia, SLR will generally reduce habitat availability, though impacts will show high spatial variability, with gently sloped shores that lack the ability to migrate landward experiencing the greatest losses of habitat. Because habitat loss is among the greatest threats to global biodiversity (Brooks et al., 2002; Mantyka-Pringle, Martin & Rhodes, 2012), it is critical to evaluate the current state of rocky intertidal ecosystems and assess how SLR may affect these important communities in coming years.

The unique ecological characteristics of rocky intertidal systems may also make them particularly vulnerable to changes in ecological structure and function as a result of SLR. The rocky intertidal zone is characterized by patterns of ecological zonation that manifest as distinct, and sometimes strict, bands along the tidal elevation gradient. These bands are generated by a variety of spatially variable and density-dependent biological mechanisms, such as competition (Connell, 1961; Menge, 1976), mutualism (Menge, 1995; Bertness & Leonard, 1997), predation (Paine, 1969; Paine, 1974; Paine, 1980; Menge, 1976), and differences in physiological tolerances to extreme environmental conditions (Connell, 1972; Paine, 1974). Regardless of the potential for the concomitant provisioning of suitable space via erosion, SLR will cause an upward shift in this banding as the current intertidal zone is submerged. It is unclear whether the changing intertidal zone will be suitable for colonization by intertidal species, as habitat characteristics, such as slope (Vaselli et al., 2008) and substrate type (Thorner, Kumar & Smith, 2014), and physical environmental conditions, such as wave stress and exposure (Harley et al., 2006), may also change. Thus, coastal squeeze and the rapid upward shift in intertidal area will likely impact the abundance, distribution, and competitive interactions of rocky intertidal species.

Approaches to quantifying climate change impacts in the rocky intertidal zone

Rocky intertidal habitats are among the most extensively studied marine ecosystems, and are well known for their rich and varied history of influential research (Ricketts & Calvin, 1939; Connell, 1961; Lewis, 1964; Paine, 1966; Paine, 1974; Paine, 1994; Dayton, 1971; Menge, 1976; Sousa, 1979; Paine & Levin, 1981; Underwood & Denley, 1984; Gaines & Roughgarden, 1985; Roughgarden, Gaines & Possingham, 1988; Underwood, Chapman & Connell, 2000, to list a few). Despite much advancement in the field, a combination of factors has constrained field sampling designs (Andrew & Mapstone, 1987). Many studies have been tradition bound, selecting sampling units a priori rather than empirically evaluating the appropriate grain size for the particular question of interest (Andrew & Mapstone, 1987; Wiens, 1989; Underwood, 2000; Denny et al., 2004). Logistical challenges, such as difficulty of access and a brief sampling window constrained to periods of emersion, have necessitated labor-intensive in-situ surveys, which are limited in spatial extent (Blakeway et al., 2003). While remote sensing approaches have been used to collect data on larger spatial scales, technological limitations have somewhat restricted the resolution and deployment of these approaches. Such limitations have previously been a barrier to determining how SLR will modify ecological patterns and processes in rocky intertidal communities at ecologically relevant spatial scales (Runting, Wilson & Rhodes, 2013).

Recently, remote-sensing approaches and geographic information system (GIS) analysis software have been used to provide high-resolution, landscape-scale ecological information in the rocky intertidal zone (Guichard, Bourget & Agnard, 2000; Blakeway et al., 2003; Bryson et al., 2013; Thorner, Kumar & Smith, 2014; Gomes et al., 2018; Konar & Iken, 2018; reviewed by Garza, 2019). Importantly, when combined with detailed in-situ information, these approaches now provide researchers the opportunity to investigate ecological dynamics at scales commensurate with the landscape processes affecting these communities (Murfitt et al., 2017; Gomes et al., 2018; Garza, 2019).

We investigated the potential ecological impacts of future SLR on rocky intertidal ecosystems, using a multi-scale approach at two marine reserves in San Diego, CA, USA as a case study. Light Detection and Ranging (LiDAR) data were used to estimate current intertidal habitat extent at four study sites and site-level habitat area changes under a range of SLR scenarios. Using high-resolution in-situ imaging, we mapped intertidal habitats and quantified the percent cover of dominant benthic space occupiers and the density of sessile and mobile invertebrates across tidal elevations in 180 m2 plots at each site. We then used a model-based approach to investigate future SLR driven changes in the areal extent of dominant benthic space occupiers and the numerical abundance of focal rocky intertidal invertebrate taxa. This work takes a critical step toward determining the future impact of SLR on rocky intertidal communities at ecologically relevant scales, and provides a framework for future monitoring and experimental efforts.

Methods

Survey locations and sites

Two rocky intertidal study locations in San Diego County were chosen: the Scripps Coastal Reserve (SCR) and the Cabrillo National Monument (CNM) (Fig. 1). Within these locations, one site was sampled at SCR (SCR 0), and three sites were sampled at CNM (CNM 1-3). These locations are recognized for their ecological and economic importance, and are both designated as marine protected areas (MPAs) under California state legislation. Regular long-term ecological monitoring by the Multi-Agency Rocky Intertidal Network (MARINe) has occurred at one site at SCR since 1997, and at three sites at CNM since 1990 (UCSC, 2020). The SCR site was studied under a permit granted by the Scripps Coastal Reserve Manager (application #: 33783), and CNM sites were studied under a permit granted by the US Department of the Interior National Park Service, Cabrillo (permit #: CABR-2016-SCI-0007).

Figure 1 Study site overview.

Location of large-area imaging plots (orange rectangles) in San Diego, CA, USA. Sites were selected to fall within long-term monitoring areas (upcoast and downcoast boundaries, black lines), and were bounded by Highest Astronomical Tide (HAT, light red contour) and Mean Lower Low Water (MLLW, dark red contour).

As previously described by MARINe, the intertidal habitat at SCR is comprised of a gently sloping boulder-field backed by a large metamorphic dike running south by southwest through the site. The dike provides a distinct upper limit to the intertidal zone, which is immediately adjacent to steep cliffs that rise vertically to an elevation of approximately 75 m. CNM2 and CNM3 are characterized by broad and gently sloping sandstone benches, while CNM1 is comprised of large scattered boulders atop the same sandstone base found at the other CNM sites. At CNM, the intertidal zone is backed by low cliffs, which rise steeply to approximately 5–10 m elevation, then more gradually to a higher ridge. Cliff stratigraphy in both locations is comprised of lithified mudstone, siltstone, shale, and sandstone, and these cliffs regularly experience failure and variable rates of net yearly erosion (Emery & Kuhn, 1980; Benumof et al., 2000; Young & Ashford, 2006).

SLR scenarios and intertidal area estimation

In the Fifth Assessment Report, the United Nations Intergovernmental Panel on Climate Change (IPCC) projected a rise in global sea-level of between 0.26 m and 0.98 m by 2100 (Church et al., 2013). More recently the National Oceanic and Atmospheric Administration (NOAA) released projections that include SLR extremes of up to 2.5 m on US coastlines (Sweet et al., 2017). Sea-level rise projections for California generally fall within the range of global projections. Cayan et al. (2008) forecasted SLR on the coast of California of 0.11–0.72 m by 2070–2099. The National Research Council (NRC) projected 0.42–1.67 m of SLR by 2100 for the coast of California south of Cape Mendocino (National Research Council, 2012). For this study, SLR scenarios from 0.10 to 2.0 m were analyzed in 10.0 cm increments (twenty scenarios) to cover the generally accepted potential range of SLR for the California coast in the next century. By analyzing SLR in increments, we also avoided using projections for specific dates, and thus our analysis is not time specific and is more flexible to the uncertainty of the projections.

The total area of the intertidal zone at each site currently, and under each SLR scenario, was estimated from the 2009–2011 California Coastal Conservancy Coastal LiDAR Project: Hydro Flattened Bare Earth Digital Elevation Model (“NOAA, 2020”), which is hereafter referred to as the site-scale DEM or SDEM. WGS1984 and MLLW were selected as horizontal and vertical datums to allow later embedding of orthophotomosaics and direct referencing to common tidal datums. Tidal datums from NOAA tide stations nearest our survey sites were selected as the initial vertical boundaries of each site (Table S1, Fig. 1). Mean Lower Low Water (MLLW, 0 m) was used as the lower boundary of the intertidal zone because it is a commonly used datum and the penetration limitations of LiDAR do not allow accurate data below MLLW. Highest astronomical tide (HAT) was chosen as the upper boundary of the intertidal zone to encompass the zone that is only covered during the highest high tides and the spray zone. Straight lines running perpendicular to the elevation contours through the northernmost and southernmost long-term monitoring plots at each site were established as fixed upcoast and downcoast boundaries (Fig. 1). The DEM Surface Tools Extension for ArcGIS (ESRI 2016. ArcGIS Desktop: Release 10.5 Redlands, CA, USA) was used to provide surface area estimates on a grid by grid basis for the SDEM (Jenness, 2004). The Surface Tools Extension accounts for vertical relief values in area estimates by using adjusted surface area values for each SDEM grid cell rather than strictly planimetric area (Jenness, 2004).

As future erosion rates are difficult to estimate, we chose to follow a passive flooding approach. In each SLR scenario at each site, total intertidal area was estimated by adjusting the vertical boundaries to the future intertidal extent (current datums + SLR), then summing surface area of SDEM grid cells within the future intertidal extent. This allowed migration of the realized available habitat upward and shoreward under each SLR scenario, so that predicted SLR was applied to the existing coastal topography of our sites in our modeling (e.g., Vaselli et al., 2008). Thus, we prevented artificial constraint on the upper limit of the intertidal zone during future scenarios. These surface area values were also subset to allow analyses by intertidal zone (e.g., lower, middle, and upper, Table S1).

Large-area imaging of intertidal survey plots and image processing

In order to characterize the composition of the biological communities at these sites, we used an imaging approach referred to as large-area imaging, which produces spatially accurate, detailed maps of the benthos from digital imagery collected in field plots (Murfitt et al., 2017; Edwards et al., 2017). The general large-area imaging workflow involves the use of commercially available Structure-from-Motion photogrammetry (SfM) software (Agisoft PhotoScan Professional V.1.3 software (Agisoft LLC 2014, St. Petersburg, Russia)). This software allows creation of composite 3-dimensional reconstructions, also known as 3D models, from raw imagery. We then generate plot-scale Digital Elevation Models (PDEM) and orthorectified 2-dimensional top-down views, or orthophotomosaics, of the 3D models (Fig. 2). These 2D models are then scaled and uploaded into ArcGIS for ecological analysis and extraction of landscape-scale metrics.

Figure 2 Survey plot orthophotomosaic and Digital Elevation Model (DEM).

(A) Example of 6 m × 30 m (180 m2) benthic landscape orthophotomosaic with zoomed in inset, and (B) plot-scale digital elevation model, for study site Cabrillo National Monument 3 (CNM 3).

At each site, imaging plots were positioned to span the maximal distance across the intertidal zone while also overlapping a subset of the long-term study sites. To establish plot locations, ten random coordinates were first generated in ArcGIS within the upper zone of the SDEM at each site. Coordinates were then ground-truthed to ensure they occurred on natural substrate within representative upper intertidal habitat. One coordinate was then randomly selected from the coordinates meeting these criteria as the upper, upcoast corner for the plot at each site. Rectangular 6.0 m × 30.0 m plots were then established running perpendicular to the shoreline along the elevation gradient. Plots of this size were deemed sufficient for accomplishing project goals while also balancing image and data processing capabilities.

Imagery was collected at all sites between December 2016 and January 2017, concurrent with the season’s lowest tides and with existing long-term monitoring surveys. A GoPro Hero 5 camera (GoPro Inc., San Mateo, CA) was mounted to a frame on a handheld transect line and passed between two surveyors across the plot, with adjacent passes separated by 0.5 m. Images were collected every 0.5 s using a linear field of view setting with an equivalent focal length of 24–49 mm. The camera was held approximately 1.0 m above the substrate to maximize overlap of images while also ensuring sufficient image resolution for benthic community identifications. Ten scale bars of known length (0.5 m) were deployed throughout the plot as x, y, z spatial references, and ground control point (GCP) coordinates were collected at the upcoast end of each scale bar. Images were collected over a 9.0 m × 33.0 m area to ensure that the target plot was imaged with sufficient coverage (1.5 m buffer around perimeter) to avoid areas missing data within the study plot.

The details of 3D model and orthophotomosaic creation have been described in detail elsewhere (Burns et al., 2015; Naughton et al., 2015; Murfitt et al., 2017). Briefly, Agisoft was used to first align imagery, estimate camera positions and scene geometry, and produce a sparse cloud of points extracted from the imagery. These sparse clouds were then optimized to correct model geometry and minimize alignment error. They were then assigned a coordinate system and scale using GCP coordinates and lengths from the scale bars within each plot. Textured dense point clouds were then produced and subsequently meshed to provide PDEMs (1.0 cm nominal post spacing). Finally, top down orthophotomosaics (0.3–0.4 mm/pixel resolution) were produced to allow visual identification and quantification of benthic organisms (Fig. 2). This resolution allowed identification of organisms of sizes greater than or equal to approximately 1.0 cm in diameter, though taxonomic resolution varied depending on morphology. Orthophotomosaics and PDEMs were exported from Agisoft in a raster format and uploaded into ArcGIS for subsequent data extraction.

Biological data extraction

Percent cover estimates

To characterize current percent cover of dominant benthic space occupiers across tidal elevations at each site, stratified random point sampling was conducted within each orthophotomosaic using ArcGIS. First, a grid of 9600 equal sized rectangular cells (12.5 cm × 15.0 cm) was generated across each plot. Then, one point was randomly placed within each grid cell, and these points were manually identified to the highest taxonomic resolution possible (see Table S2 for species identifications). Conspicuous seaweeds and invertebrates were identifiable to the species level, but morphologically indistinct (e.g., non-coralline crust algae, foliose red algae) and smaller taxa, or taxa that grow in mixed communities (e.g., articulated coralline algae, turf algae, crustose coralline algae), were grouped into functional categories. Estimates of SLR driven changes were only made for those taxa that (1) were directly attached to the benthos anywhere they were visible, and (2) were easily distinguished from other taxa in our orthophotomosaics. In situations where epiphytes were present (e.g., fleshy algae growing on mussels) we identified the taxa attached to the benthos upon which the epiphytes were growing.

Tidal elevation for each stratified random point was extracted from the PDEMs, allowing the calculation of the percent cover of each taxon in 10.0 cm tidal elevation bins (hereafter referred to simply as bins) across each plot. The percent cover of each taxon in each bin within each plot was calculated as the percent of points falling on that taxon out of the total number of points falling in the respective bin. These percent cover values were later used to estimate areal extent of dominant benthic space occupiers in each SLR scenario.

Invertebrate density estimates

To estimate the current density (#/m2) of focal invertebrate taxa across tidal elevations, systematic counts of invertebrates larger than 1.0 cm in diameter were conducted within each plot. This included both motile and sessile species (Table S2). Species counts were conducted at a consistent scale (1:4), allowing identification near the limit of orthophotomosaic resolution. Only invertebrate taxa that could be clearly identified at this scale were counted. Because this approach is image based, it is likely that taxa occupying cryptic habitat or hidden by layering, as well as smaller juveniles, were underestimated in our counts. The density of each taxon in each bin in each plot was calculated as the total number of individuals observed divided by the total area within each elevational bin as calculated from the PDEMs. These data were later used to provide estimates of numerical abundance of invertebrate species under each SLR scenario.

SLR impacts to areal extent and invertebrate abundances

In order to assess how changes in habitat area as a result of SLR may influence the rocky intertidal community at our sites, the total areal extent (m2) of dominant benthic space occupiers and the abundance of focal invertebrate taxa (#) was estimated for each site under each SLR scenario. These estimates were produced by extrapolating the estimates of percent cover and density from the orthophotomosaics and PDEMs to the more spatially expansive SDEMs. The total areal extent (m2) of each dominant benthic space occupier in each SLR scenario at each site was calculated by multiplying the percent cover of each species in each bin by the total area in the respective bin as calculated from the SDEM, and then summing across bins. This metric provided an assessment of how SLR will influence absolute areal extent of each taxon and allowed comparison of relative changes in area. The abundance for each focal invertebrate taxon in each SLR scenario at each site was calculated by multiplying the current density of each taxon in each bin by the total area in the respective bin as calculated from the SDEM, and then summing across bins.

Results

Change in intertidal habitat area and zonation with SLR

We estimated habitat area within current and future intertidal elevation ranges at our study sites and found that SLR will substantially reduce total intertidal habitat area (m2) (Fig. 3). Following a SLR trajectory consistent with the observed trend in San Diego, CA (approximately 20.0 cm by 2100), we estimate that total intertidal habitat area loss will be on average 29.88% (±3.78, SE) across study sites. Under the IPCC upper-end global projection of 1.0 m by 2100 (Church et al., 2013), this value will reach 77.72% (±4.65). Under the NRC upper-end projection for California (National Research Council, 2012) of 1.7 m, this value will rise to 85.32% (±2.33). Habitat loss will be greatest for the lower and middle intertidal zones, which currently occupy a broad and gently sloping intertidal shelf that will rapidly become subtidal as sea-levels rise (Fig. 3). Under scenarios greater than 0.2 m the lower intertidal zone will nearly always experience the greatest proportional habitat area loss, followed by the middle, then upper zones (Fig. 3). As a result, we expect that the proportional contribution of each zone to total intertidal area will shift, with the contribution of the lower intertidal zone diminishing, and that of the middle zone and the upper zone increasing (Fig. 3, Fig. S1).

Figure 3 Intertidal area changes with sea-level rise.

Site and zone level intertidal habitat area under 0–2.0 m of sea-level rise at survey sites (A) Scripps Coastal Reserve (SCR 0), (B) Cabrillo National Monument 1 (CNM 1), (C) Cabrillo National Monument 2 (CNM 2), and (D) Cabrillo National Monument 3 (CNM 3). (E) Site and zone level intertidal area change (% change) under three sea-level rise scenarios.

Change in areal extent of dominant benthic space occupiers with SLR

With little exception, we found that the SLR scenarios examined here will result in reductions in areal extent (m2) of dominant benthic space occupiers. Importantly, our data suggest that species will experience changes in areal extent of different magnitudes, resulting in changes in community structure as the relative abundance of species shifts (Fig. 4). The most pronounced changes will likely occur in the first 0.5 m of SLR for all species. At 0.5 m of SLR, we estimate a mean decrease in total areal extent across all species and study sites of 56.95% (±2.40). Changes in areal extent will be most pronounced for those taxa that primarily occupy lower and middle intertidal habitats, such as articulated coralline algae, some brown algae, red foliose algae, turf algae, and surfgrass (see Table S2 for species identifications and classifications). For example, we estimate that the total areal extent of articulated coralline algae at our study sites will decrease by an average of 83.74% (±4.72 SE; range 70.10–91.77) under the IPCC upper-end projection. Areal extent is expected to change less dramatically for taxa occupying primarily upper intertidal habitat, such as mussels and barnacles. For example, we estimate that the total areal extent of barnacles (Balanus/Chthamalus spp.) will decrease by an average of 52.99% (±12.01; range 20.38–77.84) across our study sites under the IPCC upper-end projection.

Figure 4 Sea-level rise impacts to area of benthic species.

Estimated current and future areal extent for benthic space occupiers at survey sites (A) Scripps Coastal Reserve (SCR 0), (B) Cabrillo National Monument 1 (CNM 1), (C) Cabrillo National Monument 2 (CNM 2), and (D) Cabrillo National Monument 3 (CNM 3), under 0–2.0 m of sea-level rise, showing decreases for all benthic space occupiers.

Change in invertebrate abundance with SLR

Our results suggest that there will be nearly ubiquitous declines in overall numerical abundance of sessile and mobile invertebrates at our study sites with future SLR (Fig. 5). Lower and middle intertidal taxa will likely exhibit greater population declines than upper intertidal taxa, and the largest declines will be observed in the first 0.5–1.0 meter of SLR. For example, our results suggest that the abundance of green anemones (Anthopleura sola/elegantissima) will decrease by an average of 64.37% (±8.66) and 76.20% (±15.60) across our study sites at 0.5 and 1.0 m of SLR, respectively. In contrast, we estimate smaller declines in the abundance of upper intertidal periwinkles (Littorina spp.) with 26.22% (±14.9) and 51.19% (±0.17) and upper intertidal gooseneck barnacles (Pollicipes polymerus (Sowerby, 1883)) with 29.10% (±5.60) and 47.14% (±7.54) across study sites at 0.5 and 1.0 m of SLR, respectively. Overall our results suggest mean decreases in total abundance of all invertebrates at our study sites of 55.82% (±4.25) and 66.92% (±4.09) under the IPCC and NRC projections, respectively. These results, however, do not take into account the possibility of these species becoming increasingly subtidal.

Figure 5 Sea-level rise impacts to rocky intertidal invertebrate abundances.

(A–I) Estimated current and future abundances for focal invertebrates at survey sites Scripps Coastal Reserve (SCR 0) and Cabrillo National Monument 1-3 (CNM 1-3) under 0–2.0 m sea-level rise.

Discussion

We used a multi-scale approach to investigate how predicted SLR over the next century may affect rocky intertidal communities at four sites in San Diego, CA, USA. Our results suggest that SLR will significantly reduce total rocky intertidal habitat area at our study sites. These changes will neither occur uniformly across time nor space, but rather will be most pronounced during the first meter of SLR and within the lower and middle intertidal zones. As seas rise, the broad intertidal zones at our study locations will be quickly submerged, resulting in first rapid, then more gradual, loss of intertidal habitat area. As this bench habitat is lost, the intertidal zone will likely be squeezed into the remaining habitat, most of which is more vertical in nature. While much remains unknown regarding the realized effects of SLR along the coast of southern California, our results provide an empirically based framework to investigate the potential magnitude of changes to intertidal communities as a result of SLR.

Our findings are generally consistent with those of other studies that have attempted to forecast the effects of SLR on future intertidal area and coastal profile of rocky shores. For example, Jackson & Mcilvenny (2011), Thorner, Kumar & Smith (2014), Hollenbeck, Olsen & Haig (2014), and Schaefer et al. (2020) all estimated that SLR will significantly reduce intertidal habitat area. In addition, Vaselli et al. (2008) and Jackson & Mcilvenny (2011) suggest that as seas rise, the intertidal zone will generally steepen. Habitat area is among the most significant factors limiting the growth and abundance of rocky intertidal species (Dayton, 1971). Thus, the changes in habitat area that are likely to occur due to SLR will also likely affect intertidal community structure. Our results suggest generally negative impacts to both the total areal extent of dominant benthic space occupiers and the numerical abundance of sessile and mobile invertebrates at our survey sites, even under modest SLR scenarios well within the range projected for the next century. With reduced space for colonization, and assuming densities remain similar to what they are today, substantial declines in the populations of intertidal organisms are expected.

However, it is also possible that abundances will initially remain fairly constant, leading to increased densities of both sessile and mobile species. As habitat area is compressed, the biotic interactions known to drive community structure, such as competition and predation, are likely to change (Dayton, 1971). Additionally, interactions that were otherwise rare or non-existent, due to the spatial separation between species, may become intensified or be created (Jackson & Mcilvenny, 2011). Further, because rocky intertidal organisms exhibit distinct distributions across tidal elevations and a range of life history strategies (Connell, 1972; Paine, 1974), the impact of SLR will likely be non-uniform and unpredictable across the rocky intertidal community, both taxonomically and through time. Such changes in these biotic interactions will inevitably play an important role in ultimately determining how SLR impacts this ecosystem.

While our results apply most directly to our specific study locations, rocky intertidal shores around the globe may be susceptible to coastal squeeze, as many are composed of erosion resistant benches backed by steep coastal cliffs (Emery & Kuhn, 1982; Graham, Dayton & Erlandson, 2003). The ability of SLR driven erosion to create new habitat will depend both on the rate of erosion, which must keep up with inundation, and the availability of suitable habitat, which is dependent upon the underlying stratigraphy of the eroding shoreline (Masselink et al., 2020). In the locations considered here, which are comprised of a mix of loosely compacted sedimentary materials interspersed with more durable sandstone (Young & Ashford, 2006; Emery & Kuhn, 1980), it is unclear if erosion will be able to open up new habitat. Further, habitat loss from sedimentation may also occur as a result of changes in erosional processes at adjacent locations. Ultimately, the degree to which our results are applicable to other areas will depend strongly on complex, and likely interacting, local and regional scale processes which remain poorly understood (Nicholls & Cazenave, 2010; Masselink et al., 2020).

The expanded use of in-situ image-based approaches, such as the approach used in this study, promises to increase the scope, scale, and extent over which scientists can study rocky intertidal habitats. While these approaches can increase the spatial scale of sampling relative to traditional field survey efforts, there remain limitations inherent in the use of imagery for describing intertidal communities. Any imaging approach used in the intertidal zone will be largely restricted to assessing mostly horizontal and outer surface layer organisms, thus missing cryptic taxa and understory species. Further, taxonomic resolution is limited based upon the quality and resolution of the imagery, and the choice of imaging platform should realistically be made with careful consideration of the question of interest (Konar & Iken, 2018). However, when combined with traditional in-situ sampling and collection of voucher specimens to identify cryptic taxa, the approach outlined here has the capacity to increase the scope of data that can be collected. In particular, the ability to derive spatially explicit biological and landscape metrics will allow more accurate predictions of how intertidal communities may change over time (Guichard, Bourget & Agnard, 2000; Blakeway et al., 2003; Bryson et al., 2013; Murfitt et al., 2017; Konar & Iken, 2018; Garza, 2019).

Conclusions

Here we provide a framework for evaluating the potential effects of SLR on one of the most important marine ecosystems at ecologically relevant spatial scales. At our focal study sites in San Diego, CA, we found compelling evidence that SLR will likely substantially reduce total area of intertidal habitat and the areal extent and numerical abundance of species occupying this habitat. However, to more broadly evaluate how SLR is likely to affect the rocky intertidal zone, even larger scale approaches across more study sites and regions are needed. Future studies should consider how differences in the underlying geomorphological features of study sites such as slope, habitat complexity, and erosion rates could alter SLR impacts. Future studies should also incorporate information on physical parameters known to influence spatial heterogeneity in rocky intertidal community organization that will likely evolve under global climate change, such temperature (Helmuth et al., 2002), ocean chemistry (Kroeker, Micheli & Gambi, 2013; Kroeker et al., 2016), and wave intensity (Cayan et al., 2008).

The rocky intertidal zone is one of the most accessible of marine environments and is of immense recreational, commercial, and educational value to coastal societies worldwide. These systems are likely to be substantially modified by large-magnitude global SLR on an accelerated and uncertain timeline within the next century. Our results highlight the need for ecosystem-scale evaluations in order to quantify and visualize the global change impacts that will modify the structure and function of this unique ecosystem. Similar approaches are needed more broadly for global coastlines in order to understand how to manage and mitigate impending global change impacts (Runting, Wilson & Rhodes, 2013; Murfitt et al., 2017).

Supplemental Information

Data S1 Intertidal area estimate data

Estimates of intertidal area and percent change in intertidal area (site- and zone-level) for sea-level rise scenarios from LiDAR raster SDEM.

Click here for additional data file.

Data S2 Species benthic areal extent estimate data

Site-level estimates of benthic area and percent change in benthic areal extent of dominant benthic community members for sea-level rise scenarios.

Click here for additional data file.

Data S3 Abundance estimate data

Site-level estimates of abundance of invertebrate taxa for sea-level rise scenarios.

Click here for additional data file.

Figure S1 Intertidal zone proportion estimates

Proportion of intertidal area contributed by each tidal zone (lower, middle, and upper) under 0–2.0 m of sea-level rise for survey sites (A) Scripps Coastal Reserve (SCR 0), (B) Cabrillo National Monument 1 (CNM 1), (C) Cabrillo National Monument 2 (CNM 2), and (D) Cabrillo National Monument 3 (CNM 3).

Click here for additional data file.

Table S1 Site Tidal datums from NOAA tidal stations

Tidal datums were used to segment the intertidal into zones (upper, middle, lower). Data presented in m in reference to MLLW. Upper Zone Boundaries: HAT–MHW Middle Zone Boundaries: MHW–MLW Lower Zone Boundaries: MLW–MLLW.

Click here for additional data file.

Table S2 Species Identification Metadata

Table shows species and functional group level identifications used for stratified random point counts and invertebrate counts.

Click here for additional data file.

The authors would like to acknowledge C Amir, L Bonito, D Chargualaf, and A Martinez for their assistance collecting field data, J Jones and K Lombardo for assistance with acquiring permits and accessing Cabrillo National Monument sites, J Jenness for advice on analyzing DEM data in GIS and on the use of the DEM Surface Analyst extension for ArcGIS, SA Sandin for modeling the use of large-area imaging for studying spatial ecology, E Parnell for his natural history guidance, V Petrovic for his assistance with 3D model visualization, and N Pederson for assistance with image processing.

Additional Information and Declarations

Competing Interests

Author Contributions

Field Study Permissions

Data Availability

The authors declare there are no competing interests.

Nikolas J. Kaplanis and Clinton B. Edwards conceived and designed the experiments, performed the experiments, analyzed the data, prepared figures and/or tables, authored or reviewed drafts of the paper, and approved the final draft.

Yoan Eynaud analyzed the data, authored or reviewed drafts of the paper, and approved the final draft.

Jennifer E. Smith conceived and designed the experiments, analyzed the data, authored or reviewed drafts of the paper, and approved the final draft.

The following information was supplied relating to field study approvals (i.e., approving body and any reference numbers):

Cabrillo National Monument sites were studied under a permit granted by the US Department of the Interior National Park Service, Cabrillo (permit #: CABR-2016-SCI-0007), and Scripps Coastal Reserve was studied under a permit granted by the Scripps Coastal Reserve manager (application #: 33783).

The following information was supplied regarding data availability:

All relevant data are available as Data S1–S3.

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
