# Peer review of "Future sea-level rise drives rocky intertidal habitat loss and benthic community change"

_PeerJ, doi:10.7717/peerj.9186_

## Round 0.1 · original submission · Major Revisions

There are many revisions suggested. The authors should resubmit the article after completing all reviewers' considerations.

Reviewer 1 ·

Basic reporting

The manuscript is well written and easy to read for the most part (see comments below).

Experimental design

I don't understand the formula used to calculate the densities of the invertebrates. Why is the density calculated across the total intertidal area? Please explain.

the language in the methods should be simplified and less jargon. E.g. landscape surface area? species stratified random point count?

I strongly disagree with the decision not to consider density dependence. The end result is that for some species density was increased by 2 to 5 times! Intuitively this doesn't make any sense unless there is a fundamental shift in the underlying ecology - or on a metabolic scale, a fundamental shift in the thermodynamics that would allow organisms to become more efficient and therefore be able to survive at higher densities. There is no evidence to suggest SLR will do either. Surely the better option would have been to assume that density would always be roughly similiar to present day conditions regardless of the amount of SLR?

Validity of the findings

It is unclear how the findings of this study fit within a broader context. While the models appear to show a substantial decrease in the availability of habitat for low and middle intertidal species, the true impact of this on their populations are less clear. For instance, low intertidal species may also be perfectly happy growing subtidally, so they may actually get more habitat overall if rock platforms become permanently submerged. Its not clear whether loss of habitat at the scale that the authors measured will have any meaningful impact on populations overall. While there may be habitat loss in some locales, there could be more habitat opened up at other locations - so the overall change in the amount of habitat available for a species is overall relatively unchanged. Yes this may have important implications for local populations - particularly when thinking about the scale of marine protected areas, for instance - but overall what are the impacts likely to be. I'd like to see the authors comment on the importance of scale when considering SLR impacts.

Although the authors briefly mention the possibility, too little is made of the potential for erosion and other geological processes to play a role in the amount of habitat that will be available under different SLR scenarios. Is it possible that in the region where this study was done SLR will result in larger/wider rocky platforms as rising waters erode softer materials at the base of cliffs, and leave the harder rocky platform? Perhaps some more information about the geological characteristics of the study sites may provide greater context for the non-American reader and support the implied assumption by the authors that erosion will be essentially zero under various SLR scenarios.

In the discussion, the authors do not cite any other papers that have examined the effects of SLR on marine communities. The authors need to compare and contrast their results with the results from previous studies. Are there broader/global trends? Are the impacts of SLR site specific?

Additional comments

The figure captions need to be improved. For instance the legends are not explained. What are SCR0, CNM1, CNM2, CNM3?

Through-out discussion: Redundancy is an issue e.g. "could potentially" - it makes it seem as if the authors are not convinced about what they are saying.

Reviewer 2 ·

Basic reporting

The manuscript was well written with appropriate literature references. There are some minor grammatical issues in the attached document.

Experimental design

There are a number of issues with the design that I have concerns about. If these issues cannot be clarified or resolved, the manuscript either needs to be greatly reduced with a focus on less specific patterns with clear statements of the limitations of the study or the manuscript should be rejected. The issues include:
1. The area of the rocky intertidal zone estimated in this study is the horizontal area, not taking into account the vertical relief. This needs to be clearly stated and the implications of this limitation discussed.
2. The authors limited their study to the Highest Atmospheric Tide (HAT). As I understand the study, this is a large deficiency in the design that greatly affects the conclusions of the study. It seems that if the authors are modelling a 2 m increase in SL, then they should have mapped out the area at least 2 m above the HAT. By limiting to HAT, they are artificially constraining the top end of tidal heights when modelling under future SLR scenarios. For example, at SCR, the maximum tidal height mapped out was a HAT of 2.177, thus a 0.0 to 2.177 tidal range. When modelling a 2.0 m SLR, this now constrains the intertidal zone in the future to a 0.0 to 0.177 m range (HAT is currently 2.177 - 2.0 m SLR = 0.177 max tidal height in the map, constrained on the lower end at MLLW of 0.0 m). The upper end of the upper intertidal zone will extend much further past what the authors have modelled (upper zone currently extends to 1.6 m at SCR thus a 2.0 m SLR means the upper end will extend to the current equivalent of 3.6 m, ~1.5 m above there area mapped out). The authors did not establish what is past the HAT which, at some of the sites, includes some gently sloping rocky habitat. If the habitat above the HAT is sandy beach, it is likely that the sand will be transported out of the system, exposing the rocky habitat underneath. If there is a vertical wall past the HAT, the vertical habitat will be used by intertidal organisms, although the tidal gradient will be greatly “squeezed”; but that is not included in the model. Although difficult to model, a vertical wall will also erode as the sea level rises slowly over time thus establishing more horizontal surface where tidal zones can extend into. This also was not discussed. I believe the limitations to the design (only mapping to HAT) severely reduces the value of the study and, unless a strong rebuttal is provided by the authors, suggest this paper should not be published. At minimum, in the light of the design issues, the specific conclusions of the study should be greatly reduced and the large number of limitations clearly stated.
3. I had a number of issues with the biological sampling procedure and data usage. First, the authors need to define “benthic space occupiers”. This typically describes the organisms attached to the rock. However, Egregia and Silvetia, for example, only attaches to the rock in a pretty small area but the seaweed itself can cover large areas of rock during low tide due to its length. From pictures, you can only see the seaweed canopies in these scenarios, not the benthic space occupiers that are underneath. Second, there are problems associated with the identification of species listed in S2 with species either misidentified or grouped incorrectly. See the document for a specific list. While most of the misidentifications do not necessarily alter the general patterns describe (e.g. misidentification of chitons as Katharina spp. doesn’t change the fact that a different genus of chiton will decline in the future), they do raise some concerns about the authors familiarity with the system. Third, assessments of the biological communities are limited to the horizontal surface and this should be clearly stated. Because of this, percent cover estimates are acceptable (again, assuming the limitations of horizontal surface sampling stated) but I do not believe species counts are valuable here, particularly for some of the target species for density analyses. Many of the target species are hard to count as they are in cracks and crevices, in the mussel matrix, underneath seaweeds, etc. Even in the field, counting some of these organisms is difficult; utilizing photos to make these counts are likely to be highly inaccurate. Hiding is likely to be even more evident for some of the species on the upper tidal zone edge of their tidal distribution (as they attempt to minimize desiccation stress) thus the upper tidal limits of these species measured in this study is likely incorrect.
4. I do not have a lot of faith in the accuracy of the biological data because of the issues raised above. What the authors might be better off doing is choosing a set of taxa that create clear zonal trends at the sites (on a horizontal level), probably sticking to species that tend to clump together (e.g. surfgrass, mussels, red algal turfs, rockweeds, Tetraclita barnacles, acorn barnacles) and are easy to distinguish in photos. The authors can then determine the tidal range of these populations (upper and lower tidal extents), followed by modelling similar to what the authors conducted for the different zones (high, mid, low). The authors appeared to do this with the entire species list but many of these taxa are not really limited to any particular zone. For example, Ulva, green algal turfs, Petalonia, and Phragmatopoma often establish in disturbed areas or, being opportunistic species, establish for short periods of time here are there throughout the intertidal zone (regardless of tidal height). In this case, SLR is not going to act the same on those taxa as compared to those that clearly establish within a species tidal zone. Other taxa are typically limited to pools (e.g. Sargassum, Anthopleura sola) which you cannot really model here.
5. The section on “Impacts of SLR to benthic cover”: The authors used the term “cover” through the paragraph but they are actually referring to area or aerial extent, not cover. The area was reported to decrease for numerous taxa with future SLR scenarios, not surprising given the area of the rocky intertidal as a whole decreased with SLR. By cover (%) is a different parameter and the patterns described are not necessarily true. For example, for ease of of describing the patterns, let us say that at SCR, 100% of the middle intertidal zone is rockweeds. With 45% of the site being middle intertidal zone, that means that the site is 45% rockweeds. At the 1.0 SLR scenario, the middle zone is now ~60% of the site thus a mid zone of 100% rockweeds will now cause rockweeds to make up 60% of the site in the future. Change the discussion of “cover” in this paragraph to area.

Validity of the findings

Given the issues mentioned in review of the experimental design in section 2, it is difficult to grasp how much of the conclusion are valid.

It was stated on page 18, Lines 382-383 “Our results suggest that lower and middle intertidal species will generally experience the greatest losses of benthic cover and abundance.” Assuming all of your patterns remain valid after addressing the experimental design issues, this statement is only true for the sites that the authors studied and should not be a generalized conclusion. Depending on the site slope and extent of high, mid, and low zones, a number of different outcomes could occur. For example, if the high zone is currently extremely large, this could result in an increase in the area of the lower intertidal zone with SLR.

Additional comments

I think the study is of value and, certainly, modelling of the impacts of SLR on intertidal communities needs to be further understood. However, in my opinion, there are a number of issues with the study design that I am not sure can be resolved, at least as currently described. A number of other comments not discussed above are also included in the attached document.

Annotated reviews are not available for download in order to protect the identity of reviewers who chose to remain anonymous.

---

## Round 0.2 · accepted · Accept

The manuscript is ready, but I am suggesting that the authors to make the small changes considered by the reviewers. This can be accomplished while in production - after these minor changes, the manuscript may be published.

Reviewer 1 ·

Basic reporting

Manuscript has substantially improved since previous version.

Experimental design

The methods section is substantially easier to follow than the previous version.

Ln 263: would have also missed smaller juveniles, which presumably may be more numerous than adults.

Not necessary to answer: I wonder if the authors compared the data (Ln 239-243) with the data obtained from their previous studies to see if the cover estimates corresponded.

Validity of the findings

Data has been provided.

Results section talks mostly about the extreme upper end of the different SLR scenarios. This is fine, but de-emphasises the important finding that most of the negative impacts of SLR in the actually occur within the range of the scenarios that are far more likely to be seen within the next century.

I'd like to see a bit more clarity provided in the results. For example, the authors should refer to a figure or table when reporting a result, so the reader needn't search to find the relevant data.

Ln 327: This is a little misleading as the gooseneck barnacles were only found at one site.

Additional comments

Suggest move paragraph beginning with Line 345 to later in the discussion. It would be nicer to read about the results and how they correspond with previous studies before reading about all the caveats with the design.

Reviewer 2 ·

Basic reporting

As with the original manuscript, this was well written with appropriate literature references. The modifications made in this revision have further improved the basic reporting.

Experimental design

The authors have done an excellent job in responding to reviewer comments, I appreciate the thoroughness in explaining modifications and rebuttals. It was clear from some of their responses that a couple of my original major issues with the manuscript were due to my misunderstanding of the experimental design that edits to the manuscript have fixed (thus were not critical errors as originally thought, just methods that needed further explanation).

Validity of the findings

The study is of value and the modifications to the applicability of the results elsewhere are well done, making sure to remove some overstatements present in the first submission. In addition, some of the limitations of the study are more clearly discussed but articulated in a way that do not take away from the important findings of the study.

Additional comments

Just wanted to recognize again the excellent job of the authors in responding to the reviewer comments and critiques. After reading the first version, I had strong doubts about the value of the paper. The modifications made to the text, adjustment of some of the data analyses, and revision of the discussion have completely changed my mind. Well done. I have a very few minor edits in the included Word document.

Annotated reviews are not available for download in order to protect the identity of reviewers who chose to remain anonymous.